# Power laws in pressure-induced structural change of glasses

Huijun Zhang [1], Kaiyao Qiao [1] & Yilong Han [1]✉

Many glasses exhibit fractional power law (FPL) between the mean atomic volume $v_a$ and the first diffraction peak position $q_1$, i.e. $v_a \propto q_1^{-d}$ with $d \simeq 2.5$ deviating from the space dimension $D = 3$, under compression or composition change. What structural change causes such FPL and whether the FPL and $d$ are universal remain controversial. Here our simulations show that the FPL holds in both two- and three-dimensional glasses under compression when the particle interaction has two length scales which can induce nonuniform local deformations. The exponent $d$ is not universal, but varies linearly with the deformable part of soft particles. In particular, we reveal an unexpected crossover regime with $d > D$ from crystal behavior ($d = D$) to glass behavior ($d < D$). The results are explained by two types of bond deformation. We further discover FPLs in real space from the radial distribution functions, which correspond to the FPLs in reciprocal space.

[1] Department of Physics, The Hong Kong University of Science and Technology, Clear Water Bay, Hong Kong, China. ✉email: yilong@ust.hk

G lasses are amorphous solids and ubiquitous in our daily life and in industry, but the understanding of glasses remains a major challenge in science[1,2]. In particular, microscopic structural changes in response to mechanical deformation is poorly understood[3–5]. A well-known puzzle is the fractional power law (FPL) in the reciprocal space of many metallic glasses[3,6], whose mechanism and generality remain controversial[3,6–9].

For crystals, the position of Bragg diffraction peaks is inversely proportional to the lattice plane distances in real space, i.e. $q_j \propto 1/a$, where $q_j$ is the position of the $j$th peak of structure factor $S(q)$ and $a$ is the lattice constant. Therefore the volume per atom $v_a \propto a^D \propto q_1^{-D}$ must hold for a $D$-dimensional crystal. Surprisingly, diffraction experiments for many metallic glasses show an FPL in three dimensions (3D),

$$v_a \propto q_1^{-d}, \tag{1}$$

with a fractional exponent $d \simeq 2.5 < D = 3$ under composition change[6,8] or compression[3,7,8]. Recently, power laws with large fluctuation of $d$ were observed in glasses under compression and composition change, which raises questions about the generality of the FPL[9]. Here we summarize five open questions: (1) Does the FPL generally hold in glasses? (2) Which factors affect the value of $d$? (3) What is the origin of the FPL? The anomalous FPLs have been attributed to atomic-scale fractal packing[8] and medium-range order[6], but both explanations are derived from a single state, not from a series of states as the FPL arises. Moreover, further studies of metallic glasses did not reveal a fractal structure[9]. $S(q_1)$ contains structural information spanning broad length scales in real space. It is therefore difficult to connect the FPL concerning $q_1$ in reciprocal space to certain structure changes in real space. (4) How does the FPL change from crystal behavior ($d = D$) to glass behavior ($d < D$)? It was not measureable before because glasses were usually produced from supercooled liquids instead of crystals[2]. (5) Does the FPL exist in two dimensions (2D)? This question has not been explored. Dimensionality strongly affects material properties and phase behavior. Recent studies showed that 2D and 3D glasses are fundamentally similar, but also differ in the local dynamics[10–12]. Here we try to answer these five questions by systematically changing the parameters in six systems in both 2D and 3D. To deepen our understanding of the FPLs, we measure not only the glass regime, but also the crossover to crystals.

Besides the FPL in reciprocal space, other structural power laws have also been observed in real space, e.g. based on the distances between neighbors in a granular glass[13] and the correlations of structural order parameters in supercooled liquids[14]. Whether the FPLs in reciprocal space relate to certain power laws in real space has not been explored. Here, we discover a new set of FPLs in real space that correlates with the FPLs in reciprocal space.

## Results
**Six model systems.** We perform simulations with three types of binary particles in 2D and 3D, i.e. a total of six systems: hard/hard spheres (Fig. 1a) in 2D (2DHH) and 3D (3DHH), spheres with the Weeks–Chandler–Andersen (WCA) potential[15] (Fig. 1c) in 2D (2DWCA) and 3D (3DWCA), and soft/hard spheres (Fig. 1e) in 2D (2DSH) and 3D (3DSH)[16]. The three types of pair interactions exhibit distinct deformation behavior, thus can help to identify which type of structural change gives rise to the FPL. The shoulder potential has been widely used to model metals, water, silica, micelles and colloids[17–21]. The mixtures of soft/hard particles can mimic materials whose components have different compressibilities such as alloy $Ce_{75}Al_{25}$ with soft Ce and hard Al atoms[22]. We define the packing fraction $\phi$ as the volume fraction

of hard spheres and the hard cores of soft spheres. Generally, when soft and hard particles are of the same size at low pressures, they can form crystals[16]. As the pressure increases, an increasing number of soft particles are compressed, resulting in finer-grained polycrystals and eventually glasses (Supplementary Fig. 1). Previously we resolved a sharp polycrystal-glass transition which distinguishes fine-grained polycrystals and glasses in 2DSH and 3DSH systems in ref. [16] Here we explore the crossover of the power law from crystal behavior ($d = D$) to glass behavior ($d < D$) for the first time. We systematically study the FPL by continuously tuning the fraction of soft particles $\eta$ and the softness $\lambda$.

The systems contain $N = 12,800$ disks in a square box for the 2D case and 10,000 spheres in a cubic box for the 3D case. Each state is directly compressed from a low-density liquid (see Methods). Hence states under different pressures are uncorrelated and the structural FPLs are not related to affine or non-affine deformation. As a comparison, we also compressed the 2DSH system step by step and obtained the same FPLs. The step by step compression in 3DSH system yields similar FPLs in the crystal and glass regimes, but different behaviors in the crossover regime.

**Power laws in 2D systems.** We calculate the structure factor

$$S(q = |\mathbf{q}|) = \left\langle \sum_{j=1}^{N} e^{i\mathbf{q}\cdot\mathbf{r}_j} \sum_{k=1}^{N} e^{-i\mathbf{q}\cdot\mathbf{r}_k} \right\rangle N \tag{2}$$

from the positions of particles, $\mathbf{r}$. $q_1$ is measured from the Lorentzian fit of the first peak[23].

Figure 1b, d, f shows that $v_a \propto q_1^{-d}$ holds in the 2DHH, 2DWCA and 2DSH glasses. $d \simeq D = 2$ for the 2DHH and 2DWCA systems (Fig. 1b, d), indicating a uniform deformation at all length scales like a crystal under compression. As hard disks cannot overlap, the uniform deformation in the 2DHH glass arises from squeezing the free volume of the gaps between particles. The uniform deformation in the 2DWCA glass arises from both the free-volume squeezing and the compression of WCA particles. As small and large WCA particles have the same softness characterized by the repulsive potential $U(r) \sim r^{-12}$, their size changes are proportional to each other, resulting in a uniform deformation. By contrast, the shoulder potential has two length scales which causes nonuniform deformation at the single-particle length scale and gives rise to two distinct FPLs in crossover and glass regimes (Figs. 1f and 2).

Unlike binary HH and WCA systems, SH systems can form crystals at low pressures and glasses at high pressures. Figure 1f shows three regimes of $v_a \propto q_1^{-d}$ in the 2DSH system. In the crystal regime, $d = 1.95 \pm 0.05 \simeq D = 2$, as expected. In the glass regime, $d = 1.36 < 2$, similar to the FPLs with $d < 3$ in 3D metallic glasses[3,6]. We find that the crossover regime can also be fitted by an FPL with $d = 4.6$ (Fig. 1f).

The FPL in Fig. 1f is replotted in Fig. 2 as a function of packing fraction $\phi$,

$$q_1 \propto \phi^{1/d}, \tag{3}$$

in order to compare the three power-law regimes with the five regimes observed in refs. [16]: 1 polycrystals at $\phi < 0.66$ featuring Hall–Petch behavior, i.e. the mechanical strength increases as the crystalline grains become finer; 2 ultrafine-grained polycrystals at $0.66 < \phi < 0.70$ featuring inverse-Hall–Petch behavior[24]; 3 shadow glass at $0.70 < \phi < 0.76$ featuring strong dynamics[25]; 4 low-density glass at $0.76 < \phi < 0.80$; and 5 high-density glass at $\phi > 0.80$. The boundaries of these five regimes were identified from various structural, dynamic, mechanical and thermodynamic quantities[16]. Here the FPL provides new features at their boundaries: the three power laws in Fig. 2 intersect at the boundary between the Hall–Petch and inverse-Hall–Petch regimes and the boundary

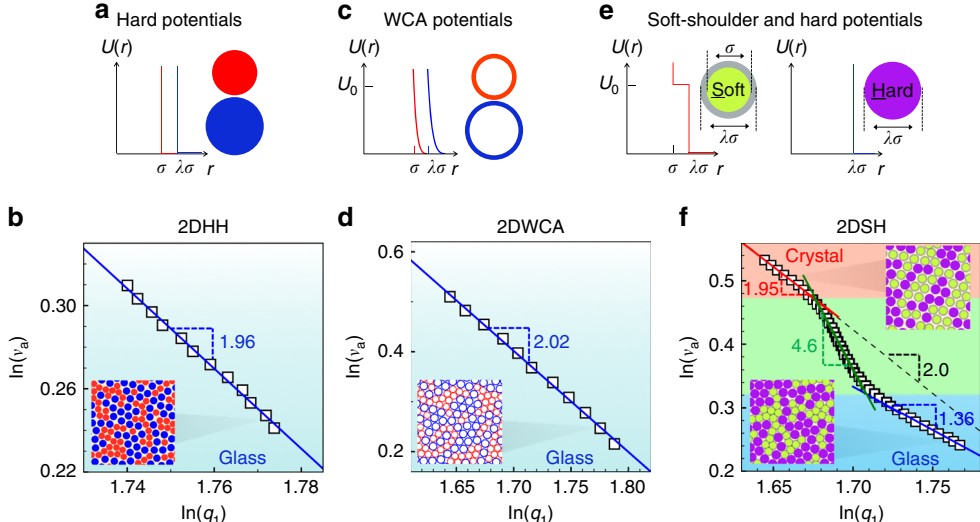

**Fig. 1** $v_a \propto q_1^{-d}$ **of three binary systems with mixing ratio and size ratio** $(\eta, \lambda) = $**(0.5, 1.3).** (**a**, **c**) Hard and WCA potentials. (**b**, **d**) $v_a \propto q_1^{-d}$ with $d \simeq 2.0$ (blue lines) in 2DHH and 2DWCA glasses, respectively. Insets are part of the glass configurations. **e** The interaction potentials of square-shoulder and hard spheres. The diameter of the inner core of the soft particle, $\sigma$, is the unit length. The outer shell diameter $\lambda\sigma$ is the same as that of hard disks. $U_0$ is the unit energy. **f** Compression-induced crystal-glass transition in the 2DSH system exhibits three regimes: power law in the polycrystal regime with $d = 1.95$ (red line), FPL in the crossover regime with $d = 4.6$ (green line), and FPL in the glass regime with $d = 1.36$ (blue line). The black-dashed reference line $v_a = q_1^{-2}$ is for a perfect crystal. Insets are typical configurations of the crystal at $\phi = 0.64$ and the glass at $\phi = 0.82$ (see Supplementary Fig. 1 for the full views of more states).

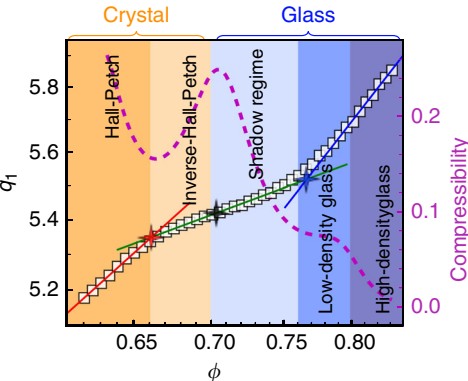

**Fig. 2 Five regimes of the 2DSH system with** $(\eta, \lambda) = $**(0.5, 1.3).** Their real-space structures are shown in Supplementary Fig. 1. Red, green and blue lines in the log-log plot denote the fitted $q_1 \propto \phi^{1/d}$ in crystal, crossover and glass regimes, respectively. The intersection of the three lines coincide with the boundary between Hall–Petch and inverse-Hall–Petch behaviors (red star) and the boundary between the shadow glass and normal glass regimes[16]. The slope of $q_1(\phi)$ reaches the minimum at $\phi = 0.70$ (black star), coinciding with the polycrystal-glass boundary identified in ref. [16]. The five regimes are identified in ref. [16] and can be roughly seen from the compressibility (dashed curve).

between the shadow and low-density glasses. In addition, the minimum slope of $q_1(\phi)$ (black star in Fig. 2) coincides with the boundary between the polycrystal and glass regimes identified via other methods[16]. These results generally hold in other 2DSH systems with different values of $(\eta, \lambda)$. Hence the $q_1(\phi)$ curve could provide empirical criteria to distinguish between Hall–Petch and inverse-Hall–Petch regimes, and between ultrafine-grained polycrystal and glass regimes, at least in 2DSH systems.

The FPL in the crossover corresponds to a regime with abnormally large compressibility (dashed curve in Fig. 2). The low- and high-density glasses in Fig. 2 have been observed[16,26]

when the particle interaction has two length scales such as the square-shoulder potential. Here we find that low- and high-density glasses have the same $d = 1.36$ (Fig. 2), which is consistent with the observation that both the low- and high-density metallic glasses of $Ce_{68}Al_{10}Cu_{20}Co_2$ have the same FPL with $d = 2.5$[3].

**Fraction of soft shells governing the FPL.** 2DSH systems with different values of $\eta$ or $\lambda$ similarly exhibit three power laws at the crystal, crossover and glass regimes as shown in Fig. 3a, b, respectively. The exponent $d$ in the glass regime varies with $\eta$ and $\lambda$ (Supplementary Fig. 2a, b). Interestingly, $d$ decreases linearly with the area fraction of the soft deformable part in the total area of all the particles: $X = \eta(\lambda^2 - 1)/\lambda^2$ (Fig. 3c). $X$ describes the amount of size mismatch available in the 2DSH system under compression, which determines the amount of defects that can be produced in the crystal and reflects the glass-forming ability of 2DSH crystals. We further measure three other systems, and all of their $d$ values lie on the linear $d(X)$ as shown in Fig. 3c. Therefore, we conclude that the soft deformable part governs $d$. A larger area fraction of the soft part can produce more nonuniform deformation under compression, thus $d$ deviates more from $D$ (Fig. 3c). Compression-induced FPLs have been measured in two types of metallic glasses based on La and Ce, e.g. $La_{62}Al_{14}Cu_{11.7}Ag_{2.3}Ni_5Co_5$ and $Ce_{68}Al_{10}Cu_{20}Co_2$. They both yield $d \simeq$ 2.5[3,7], indicating that they have similar fractions of soft compressible parts. In metallic glasses, Al, Cu, Ag, Ni and Co are known to be hard-sphere-like atoms, while La and Ce are much softer due to their localized electrons[22,27]. In fact, Ce can be described by the square-shoulder potential[17].

Extrapolating $d$ to 2 and 1 gives $X = 0.128$ and 0.246, respectively (Fig. 3c). These values correspond to $\eta = 0.314$ and 0.603 when $\lambda = 1.3$ (Fig. 3a), and $\lambda = 1.16$ and 1.40 when $\eta = 0.5$. They coincide with the glass-forming regimes $0.30 \leq \eta \leq 0.60$ for $\lambda = 1.3$ and $\lambda \geq 1.16$ for $\eta = 0.5$ observed in 2DSH systems[16]. Beyond these regimes, systems either resemble a monodispersed system (e.g. $\eta < 0.30$ or $>0.60$ at $\lambda = 1.3$; $\lambda < 1.16$ at $\eta = 0.5$), or becomes a binary system with a large size ratio (e.g. $\lambda > 1.4$ at

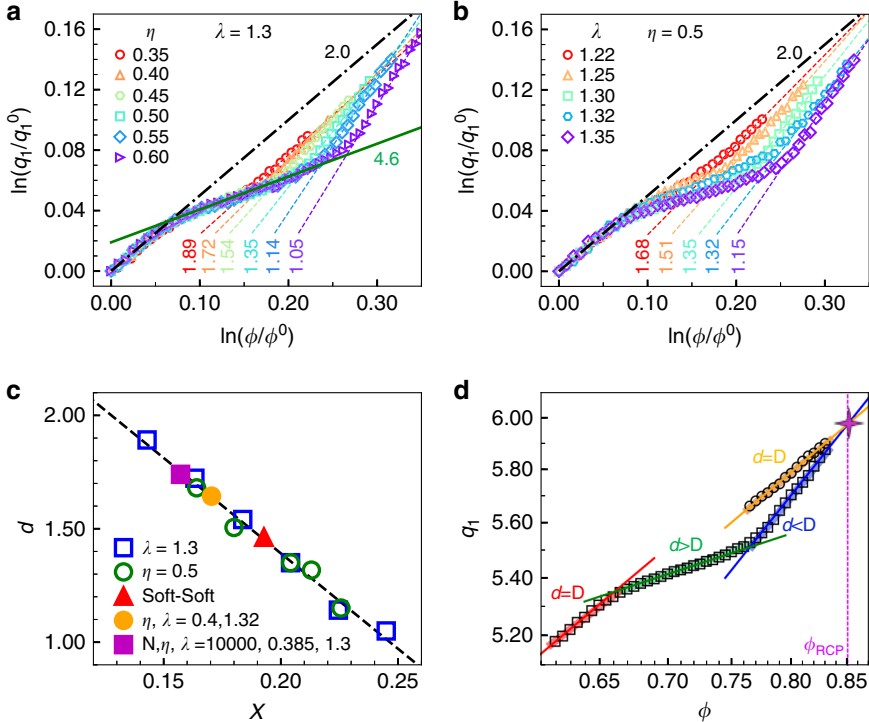

**Fig. 3 Effect of deformable part on the FPLs in 2DSH glasses. a** FPLs for systems with different values of $\eta$ at the fixed $\lambda = 1.3$. The fitted exponent $1.05 \leq d \leq 1.89$ in the glass regime and remains a constant $d = 4.6$ in the crossover regime. **b** FPLs for systems with different values of $\lambda$ at the fixed $\eta = 0.5$. **c** All $d$'s fitted from **a** and **b** collapse into a linear function of the area fraction of soft shells, $X$. We further measured three other systems and all of their $d$'s lie on this line, including the 2DSH system with $(\eta, \lambda) = (0.4, 1.32)$ (orange circles), a smaller system with $(N, \eta, \lambda) = (10000, 0.385, 1.3)$ (purple squares) and a soft/soft mixture with $(\sigma_1, \lambda_1, \sigma_2, \lambda_2) = (1, 1.28, 1.24, 1.03)$ at $\eta_1 = 0.4$ (red triangles). **d** The extrapolations of the fitted power laws of the 2DSH glass (blue line; Fig. 1f) and the 2DHH glass (orange line; Fig. 1b) meet at the 2D random-close-packing point: $\phi_{RCP} \simeq 0.850$ (purple star). Both systems have $(\eta, \lambda) = (0.5, 1.3)$.

$\eta = 0.5$), which leads to phase separation[28]. Consequently, systems beyond these regimes can only form large-grained polycrystals instead of glasses even under the highest pressure[16], and the corresponding FPL does not exist in the glass regime (Supplementary Fig. 3a, b).

**FPLs near random-close packing.** At the high-pressure limit, almost all soft particles would be compressed so that the 2DSH system becomes a 2DHH system with the same $(\eta, \lambda)$ at the random-close packing (RCP) point $\phi_{RCP}$[29]. This is confirmed in systems with various $(\eta, \lambda)$ values. For example, the extrapolations of $q_1 (\phi)$ of 2DSH and 2DHH systems with the same $(\eta, \lambda) = (0.5, 1.3)$ intersect at $\phi = 0.850 \simeq \phi_{RCP} \simeq 0.848$ of binary hard disks with diameter ratio 1.4[29] (Fig. 3d). To approach $\phi_{RCP}$, the rapid increase in $q_1 (\phi)$ (i.e. $d < D$) in the glass regime needs to be compensated by a slow increase in $q_1 (\phi)$ (i.e. $d > D$) in the crossover regime. More compressible parts (i.e. larger $\eta$ or $\lambda$) create a broader crossover regime with a stronger deviation from the line of $d = 2$, hence a steeper $q_1 (\phi)$ (i.e. smaller $d$) in the glass regime is needed for the compensation as shown in Fig. 3a.

**Power laws in 3D systems.** Similar power laws are observed in 3D systems (Fig. 4), which further confirms that the FPL, $d < D$, requires two length scales in the potential. 3DHH and 3DWCA glasses exhibit the normal power laws with $d = 3.0 = D$ (Fig. 4a, b) similar to their 2D counterparts (Fig. 1b, d). Similar to its 2D counterpart, the 3DSH system also exhibits the crystal regime with $d = 3.0 = D$, the crossover regime with $d = 4.03 > D$ and the glass regime with $d = 2.48 < D$ as shown in Fig. 4c. In contrast to the continuous $v_a (q_1)$ curve in the 2DSH system (Fig. 1f), $v_a (q_1)$

in the 3DSH system abruptly jumps at $\phi = 0.5$ in Fig. 4c, coinciding with the crystal-glass transition point (Fig. 4d). This is in accordance with the observations in ref. [16] that the crystal-glass transition is like first order in 3D and more continuous in 2D. If the 3DSH system is compressed step by step, the crystal behavior of the power law extends to the crossover regime and exhibits a jump at the onset of the glass regime. Such protocol dependence in the crossover regime should be due to the first-order-like polycrystal-glass transition in 3D.

**Theoretical explanation for the FPL.** At thermal equilibrium, the Helmholtz free energy $F = U - TS$ is minimized. $U$ is the internal energy and $S$ is the entropy related to the free volume[30]. For SH systems, when a SS or SH-bond is compressed (Fig. 5a), $\Delta U = U_0$ and $T\Delta S \propto TA_{SS}$ or $TA_{SH}$. Note that the configurational entropy is neglected because we focus on the entropy change instead of entropy when a bond is compressed in a given glassy configuration. Apparently, the compressed area $A_{SS} > A_{SH}$ as shown in Fig. 5a. Therefore, SS bonds will be compressed first as they reduce $F$ more effectively. Consequently, we expect three regimes. At low pressures, the compressed volumes come from the gaps between particles. Few SS or SH bonds have been compressed so that the structure remains a crystal. At medium pressures, the volume change mainly arises from the compressed SS bonds, resulting in a more disordered structure. The nonuniform spatial distribution of the compressed volume (Supplementary Fig. 4a) causes $d$ to deviate from $D$. At high pressures, almost all SS bonds have been compressed so that further volume shrinkage is caused solely by the compression of SH bonds. These compressions do not occur randomly in space, but only in the previously uncompressed areas (Supplementary Fig. 4b). This compensates

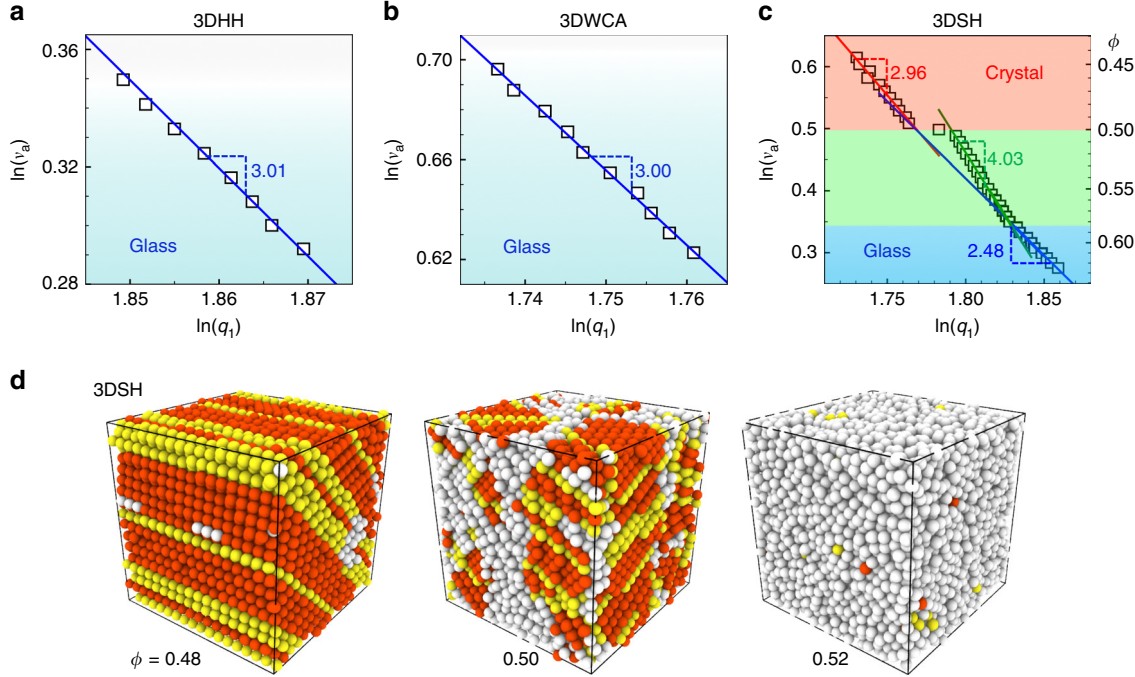

**Fig. 4 $v_a \propto q_1^{-d}$ in the 3D systems.** $v_a \propto q_1^{-d}$ (**a, b**) $v_a \sim q_1^{-d}$ with $d \simeq 3.0$ in 3DHH and 3DWCA glasses with $(\eta, \lambda) = (0.5, 1.3)$. **c** $v_a \sim q_1^{-d}$ holds in the crystal regime with $d = 2.95 \simeq D$ (red line) and in the glass regime with $d = 2.48$ (blue line). Most of the crossover regime can be fitted by $v_a \propto q_1^{-4.03}$ (green line). The abrupt jump of $q_1$ at $\phi = 0.5$ reflects a first-order-like transition in 3D. **d** Compression-induced crystal-glass transition in the 3DSH system with $(\eta, \lambda) = (0.4, 1.25)$. Orange, yellow and white spheres denote face-centered cubic, hexagonal close-packed and disordered particles, respectively. They are identified from the weighted bond-orientational order parameters (see Methods)[42,44]. The ordered structure becomes completely disordered in a small density range, $0.48 < \phi < 0.52$, reflecting a sharp transition at $\phi = 0.5$.

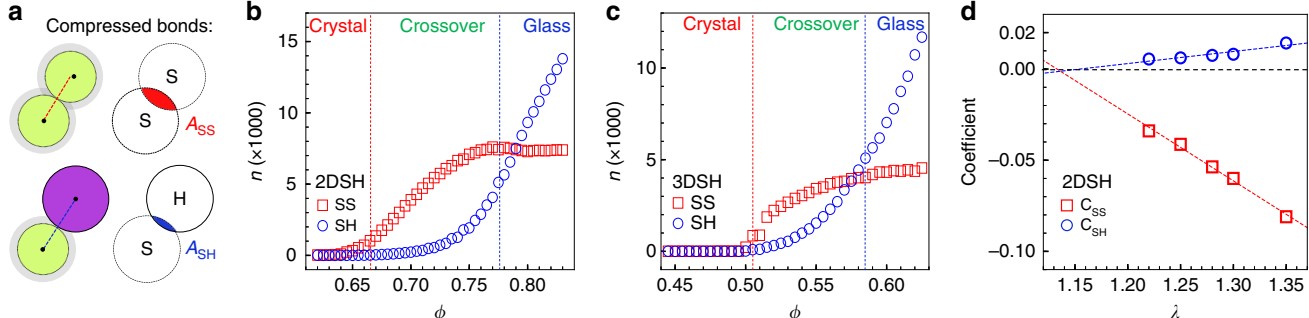

**Fig. 5 The three regimes arise from the compression of soft–soft and soft–hard bonds.** **a** The area change of compressing a soft–soft bond (red area $A_{SS}$) is greater than that of compressing a soft–hard bond (blue area $A_{SH}$). The compressed SS and SH bonds are defined as pairs belonging to the first and second subpeaks of the first-layer neighbor separations, respectively, see Supplementary Fig. 5c. **b, c** The 2DSH system with $(\eta, \lambda) = (0.5, 1.3)$ and the 3DSH system with $(0.4, 1.25)$, respectively. The number of compressed SS bonds and SH bonds shows three regimes corresponding to the crystal, crossover and glass regimes. The vertical dashed lines denote the boundaries of the crossover regime measured from Fig. 2 for the 2DSH system and from Fig. 4c for the 3DSH system. The standard deviations are smaller than the symbol size and thus the error bars are not shown. **d** The coefficients of the SS and SH terms in Eq. 5 fitted from Supplementary Fig. 7. Their linear extrapolations reach 0 (horizontal line) at $\lambda \simeq 1.14$, which coincides with $\lambda = 1.16$ when $d = 2.0$ in Fig. 3c.

for the deviation from $D$ in crossover regime, consistent with the result based on the RCP point in Fig. 3d.

The three stages are confirmed in Fig. 5b for the 2D case and Fig. 5c for the 3D case. Interestingly, the three stages in Fig. 5b, c coincide well with the crystal, crossover and glass regimes in Figs. 2 and 4c, suggesting that the compressions of SS and SH bonds are responsible for the crossover and glass regimes, respectively. Figure 5b for the 2DSH system shows that the crossover regime is dominated by the compression of SS bonds, and the glass regime is dominated by the compression of SH bonds. Figure 5c for the 3DSH system shows an abrupt increase in the number of compressed SS bonds at $\phi = 0.5$, which

coincides well with the sharp first-order-like crystal-glass transition identified in Fig. 4d. In the crossover regime ($0.50 < \phi < 0.58$), both SS and SH bonds are compressed (Fig. 5c), and their mixing effects result in a nearly power law in Fig. 4c. In the glass regime ($\phi > 0.58$), the number of compressed SS bonds is saturated so that the deformation solely arises from the compression of SH bonds (Fig. 5c). Therefore, glass and crossover regimes have distinct FPLs.

As the crossover regime is dominated by the compression of SS bonds (Fig. 5b), systems with the same $\lambda$ and different values of $\eta$ should have the same $d$ because the compression of one SS bond has the same impact on $\Delta\phi$ and $\Delta q_1$. This is confirmed in Fig. 3a.

For systems with different values of $\lambda$, compressing a SS bond changes $\phi$ and $q_1$ differently. Hence, the power laws in the crossover regime show a different $d$, as confirmed in Fig. 3b. Systems with a larger $\eta$ have more SS bonds which gives rise to a broader crossover regime (Fig. 3a). The large exponent, i.e. $d > D$, implies that $q_1 (\phi)$ increases much more slowly than it does in crystals. This can be understood as the compressed SS bonds reducing $\phi$ effectively, but affecting $q_1$ much less as $q_1$ is from the average of all bonds. Note that the compressed SS bonds at the highest pressure represent <20% of all bonds. In addition, a larger $\lambda$ gives a larger $A_{SS}$, i.e. more change in $\phi$ than $q_1$, therefore yielding a larger $d$ (Fig. 3b) in the crossover regime. For example, suppose that compressing one large-$\lambda$ SS bond and two small-$\lambda$ SS bonds results in the same $\Delta\phi$, but the structural deformation in the former case is more localized in real space and thus less effective in changing $q_1$. As this effect gives $d > D$ in the crossover regime, the glass regime should have $d < D$ for the compensation to be able to reach the RCP point.

The above arguments about the SS- and SH-bond compressions without using any simulation results have qualitatively explained $d > D$ in the crossover regime and $d < D$ in the glass regime, and $d(\lambda)$ behaviors. The simulation results in this section just provide consistency checks and are not necessary for the theoretical explanations. Beside the above approach, next we introduce another approach based on Eqs. 4 and 5 below. This second approach needs simulation results to fully explain the observations. Thus the first approach above provides a full qualitative explanation, while the second approach below is just a consistency check.

We further estimate the effects of SS and SH bonds on the FPL. The FPL indicates that $1/d$ is the slope of $\ln\tilde{q}_1/\ln\tilde{\phi}$. $\tilde{q}_1 = q_1/q_1^0$, $\tilde{\phi} = \phi/\phi^0$ and $q_1^0$ and $\phi^0$ are values of the initial single crystal (Fig. 3a, b). Under compression, $\tilde{q}_1$ is a function of the volume change. In 2DSH systems, the volume change arises from the squeezing of the free volume characterized by $\tilde{\phi}$, and the compression of the SS and SH bonds. Note that the compressed volume from a SS or a SH-bond, i.e. $A_{SS}$ or $A_{SH}$, is a constant at a fixed $\lambda$ (Fig. 5a), and thus their numbers, $n_{SS}$ and $n_{SH}$ as functions of $\tilde{\phi}$, determine the amount of volume change. Consequently, $\tilde{q}_1 = \tilde{q}_1(\tilde{\phi}, n_{SS}, n_{SH})$, and the FPL becomes

$$\frac{1}{d} = \frac{d\ln\tilde{q}_1}{d\ln\tilde{\phi}} = \frac{\partial\ln\tilde{q}_1}{\partial\ln\tilde{\phi}}\bigg|_{n_{SS},n_{SH}} + \frac{\partial\ln\tilde{q}_1}{\partial n_{SS}}\bigg|_{\tilde{\phi},n_{SH}}\frac{\partial n_{SS}}{\partial\ln\tilde{\phi}}$$
$$+ \frac{\partial\ln\tilde{q}_1}{\partial n_{SH}}\bigg|_{\tilde{\phi},n_{SS}}\frac{\partial n_{SH}}{\partial\ln\tilde{\phi}}. \quad (4)$$

The constant $n_{SS}$ and $n_{SH}$ in the first term describe the fixed numbers of SS and SH bonds. Hence the compression solely occurs from the free-volume change, which is similar to the uniform compression of SH crystals or HH glasses. Thus, $\frac{1}{d} = \frac{\partial\ln\tilde{q}_1}{\partial\ln\tilde{\phi}}\big|_{n_{SS},n_{SH}} = \frac{1}{D}$. The second and third terms denote the contributions from SS and SH bonds, respectively. $\ln\tilde{q}_1$ is proportional to $n_{SS}$ and $n_{SH}$ in Supplementary Fig. 6a, and we find a similar relationship in $Cu_cZr_{1-c}$ using the data in ref. [31] (Supplementary Fig. 6b). Consequently, $\frac{\partial\ln\tilde{q}_1}{\partial n_{SS}}\big|_{\tilde{\phi},n_{SH}}$ and $\frac{\partial\ln\tilde{q}_1}{\partial n_{SH}}\big|_{\tilde{\phi},n_{SS}}$ are denoted by two constants $C_{SS}$ and $C_{SH}$, respectively. $\frac{\partial n_{SS}}{\partial\ln\tilde{\phi}}\big|_{n_{SH}} = \frac{dn_{SS}}{d\ln\tilde{\phi}}\big|_{n_{SH}}$ and $\frac{\partial n_{SH}}{\partial\ln\tilde{\phi}}\big|_{n_{SS}} = \frac{dn_{SH}}{d\ln\tilde{\phi}}\big|_{n_{SS}}$, as $n_{SS}$ and $n_{SH}$ only depend on $\phi$. Thus, the integration of Eq. 4 yields

$$\ln\tilde{q}_1 = \frac{1}{D}\ln\tilde{\phi} + C_{SS}n_{SS} + C_{SH}n_{SH}. \quad (5)$$

Fitting $\ln\tilde{q}_1(\ln\tilde{\phi})$ curves with Eq. 5 (Supplementary Fig. 7) yields $C_{SS} < 0$ and $C_{SH} > 0$ (Fig. 5d), indicating that the

compressed SS and SH bonds increase and decrease $d$ relative to $D$, respectively. This is consistent with $d > D$ in the crossover regime dominated by compressing SS bonds and $d < D$ in the glasses regime dominated by compressing SH bonds (Figs. 3b and 5b). Moreover, $C_{SS}$ and $C_{SH}$ vanish at $\lambda \simeq 1.14$ (Fig. 5d), indicating $d \to D$ as $\lambda$ decreases toward 1.14. This is consistent with the fact that the 2DSH system cannot be compressed into a glass at $\lambda \leq 1.16$[16].

**Power laws in real-space $g(r)$.** Power laws have been observed in real-space structures of amorphous states, e.g. the $j$th pair distance $r_j (\phi)$ in a granular system[13] and correlation functions of structural order parameters in supercooled liquids[14]. These power laws cast important light on the disordered structures, but are not directly related to the radial distribution function

$$g(r = |\mathbf{r}|) = V/N\left\langle\sum_{i\neq j}\delta[\mathbf{r} - (\mathbf{r}_i - \mathbf{r}_j)]\right\rangle, \quad (6)$$

where $V$ is the volume. $g(r)$ is usually derived from the Fourier transformation of the measured $S(q)$ in scattering experiments[6]. Thus it has been used to explore the structural origin of the FPL in reciprocal space[6,8]. The FPL in reciprocal space has been attributed to the fractal structures at the length scale of the nearest neighbors, i.e. the first peak of $g(r)$[8], but ref. [9] pointed out that the fractal structure is absent at the atomic scale. Ref. [6] suggested that the FPL arises from the medium-range order from the fit of the envelop of $|g(r) - 1| \sim r^{-\gamma}\exp(-r/\xi)$. However, this fit only captures the structure at a fixed $\phi$ rather than the structural change at a series of $\phi$ values as the way that the FPL in reciprocal space is derived.

Here we discover FPLs from the $g(r)$ peaks of glasses with a series of $\phi$ values. For binary systems, the first peak of $g(r)$ splits into three subpeaks because the first-layer neighbors have three typical separations corresponding to S–S, S–H and H–H bonds (Supplementary Fig. 5c). These three subpeak positions cannot deviate much from their corresponding bond lengths, thus $r_1$ changes very little, resulting exceptionally large exponents in the power law (Supplementary Fig. 8). The second- and third-layer peaks of $g(r)$ split into more subpeaks which interferences with each other, thus their ambiguous peak positions are not measured. We focus on the positions of the unimodal peaks, i.e. fourth to eighth peaks for 2DSH glasses (Fig. 6e), third to seventh peaks for 2DHH glasses (Fig. 6a), and second to sixth peaks for 2DWCA glasses (Fig. 6c). We find that

$$v_a \propto \phi^{-1} \propto r_j^{d_j} \quad (7)$$

with $d_j = D$ for 2DHH and 2DWCA glasses (Fig. 6b, d) and $d_j < D$ for 2DSH glasses (Fig. 6f). Such a shift of the $j$th peak position, $r_j$, demonstrates that the medium-range pair distance changes uniformly with $\phi$ for 2DHH and 2DWCA glasses and nonuniformly for 2DSH glasses, in accordance with the FPLs in reciprocal space. These results suggest that the power laws in real space and reciprocal space have the same structural origin. Whether $d_j$ from real space and $d$ from reciprocal space have a quantitative relation is worth to explore in the future.

## Discussion

From the simulations of the six model systems, we found answers to the five questions about the FPL in reciprocal space raised in the Introduction section as follows:

Does the FPL generally hold in glasses? Yes, but the exponent $d$ is not a universal constant. We observed the compression-induced FPLs with $d < D$ in 2DSH and 3DSH glasses, and power laws with $d = D$ in 2DHH, 3DHH, 2DWCA and 3DWCA glasses.

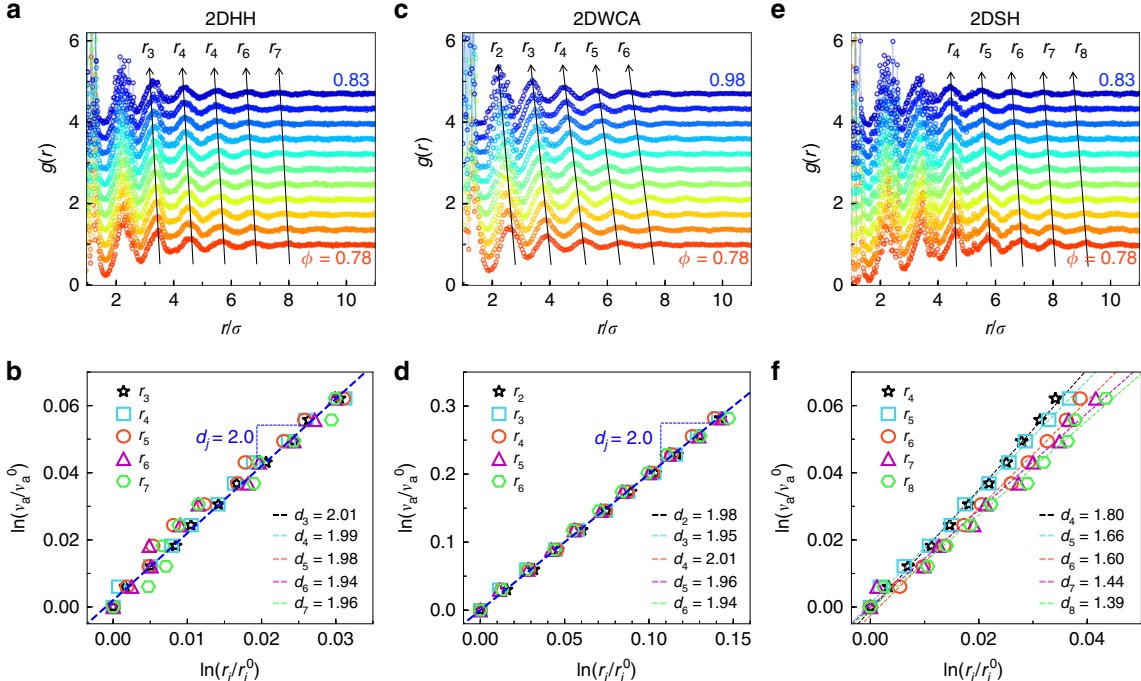

**Fig. 6 Power laws $v_a \propto r_j^{d_j}$ from the medium-range real-space structures of the glasses. a, c, e** Radial distribution functions $g(r)$ for 2DHH, 2DWCA and 2DSH glasses with $(\eta, \lambda) = (0.5, 1.3)$, respectively. The curves are vertically shifted for clarity. The arrows show the peak positions in the medium range. **b, d, f** $v_a \propto r_j^{d_j}$ measured from **a, c** and **e**, respectively. The scaling parameters $v_a^0$ and $r_j^0$ are for the highest-density glass. The exponents $d_j < D = 2$ for 2DSH glasses and $d_j \simeq D$ for 2DHH and 2DWCA glasses.

Compression or composition change can produce the power law $v_a \propto q_1^{-d}$ in our simulations and in the literature, but they have different impacts on the value of $d$. Therefore the data points with both pressure and composition change do not exhibit a good power law[9].

Which factor affects the value of $d$? The answer was previously unclear because the responsible parameters were not systematically varied. We found that the compression-induced FPL with $d \neq D$ requires a mixture of soft and hard particles so that the local structure can be deformed nonuniformly. By systematically adjusting $\eta$ and $\lambda$ of 2DSH systems, we found that $d$ is linearly governed by the fraction of the soft shells $X$ (Fig. 3c).

For particles of different sizes but with the same softness (e.g. binary WCA spheres or binary hard spheres), the interparticle distances change uniformly under compression, resulting in the trivial $d = D$. This is consistent with the observed $d \simeq 3.0 = D$ when $Zr_{46}Cu_{54}$ is compressed[9] because both Zr and Cu atoms are like hard spheres[31,32]. Note that this does not conflict with $d \simeq 2.3 < D = 3$ in $Zr_xCu_{1-x}$ metallic glasses when the mixing ratio $x$ is varied[6] because changing $x$ is analogous to compressing a SH system rather than an HH system.

The FPL was mainly observed in metallic glasses when the mixing ratio of different types of atoms was varied[6]. A few studies showed that compression can also induce the FPL with $d \simeq 2.5$ in La- or Ce-based metallic glasses, e.g. $Ce_{68}Al_{10}Cu_{20}Co_2$[3,7], but these coincidences at 2.5 do not mean that changing the pressure or changing the mixing ratio would have the same effect on the FPL. Ce atoms are much softer than other atoms[22,27] and can be described as spheres with a square-shoulder potential[17] because its 4f-electron orbit is localized at low pressure and delocalized at high pressure, resulting in atomic volume collapse[3,22], thus the metallic glasses in ref. [3] are similar to our SH systems and exhibit the FPL. However, our simulation suggests that $d$ would not be constant at 2.5 when metallic glasses with different fractions of soft atoms like Ce are compressed.

What is the origin of the FPL? We found that the FPL and its exponent $d$ are determined by different types of volume changes. For HH systems, the compression arises solely from the squeezing of the free volume, i.e. the gaps between particles. Its $d = D$ indicates that such a volume change is uniform. For WCA systems, the volume change arises from both the squeezing of the free volume and the compression of the particles. The latter is also uniform as both large and small WCA particles have similar softness, hence $d = D$ in WCA systems. For 2DSH and 3DSH systems, the volume change arises from the squeezing of the free volume and the compression of SS and SH bonds. In the crystal regime, the uniform free-volume change dominates and thus $d = D$. In the crossover and glass regimes, the volume change is dominated by the compression of SS and SH bonds, respectively (Fig. 5). Compressing a large-$\lambda$ SS bond is equivalent to compressing multiple small-$\lambda$ SS bonds in changing $\Delta\phi$, but the former deformation is more localized in space which is less effective at changing $S(q)$ at a small $q_1$. Therefore, $d$ is larger for a large-$\lambda$ system in the crossover regime. When $\lambda = 1$, the SH system reduces to the HH system where $d = D$. Hence, $d > D$ for $\lambda > 1$ in the crossover regime reflecting a nonuniform local structural change.

The volume change in the glass regime must occur in previously uncompressed local regions rather than at random positions. In other words, the structural change in the glass regime will compensate for the nonuniform structural built up in the crossover regime. Consequently, $d > D$ in the crossover regime is accompanied by a $d < D$ in the glass regime, in accordance with the same RCP structure of 2DSH and 2DHH systems. This explanation for $d < D$ and $d > D$ in different regimes does not need the assumption of any fractal structure. In fact, fractal structures with dimension $d > D$ cannot exist in a $D$-dimensional space.

Note that the FPL is not related to affine or non-affine deformation because each solid state is directly compressed from

a low-density liquid, i.e. structures at different values of $\phi$ are uncorrelated. This is also supported by the existence of the FPL in metallic glasses when their compositions change, which is not related to any non-affine deformation.

How does the FPL change from crystal behavior ($d = D$) to glass behavior ($d < D$)? We studied this question for the first time by creating a novel crystal-to-glass transformation. We found that the crossover regime between a crystal and a glass can also be fitted by a power law, but its $d > D$ does not sit between the $d$ values of the crystal and the glass. This is because the volume change in the crossover regime mainly comes from the compression of SS bonds, which is more effective at changing $\phi$ than changing $q_1$ as explained in the answer to question 2. We discovered that the onset of the crossover FPL regime coincides with the boundary between Hall–Petch and inverse-Hall–Petch regimes, while the minimum slope coincides with the polycrystal-glass transition in 2DSH systems (Fig. 2). In 3DSH systems, however, the crystal regime with $d = D$ terminates at the crystal-glass transition (Fig. 4c). These coincidences generally hold for systems with different values of $\eta$ and $\lambda$, which cast new light on the poorly understood Hall–Petch-to-inverse-Hall–Petch transition[16,24] and the polycrystal-to-glass transition[16].

Does the FPL exist in 2D? Yes, the FPL in 2D is similar to that in 3D and has been explained above. Low-dimensional systems are much softer because there are more long-wavelength fluctuations[33] and particles have fewer neighbors providing constraints. Consequently, the space dimension could strongly affect the nature of phase transition[33,34]. For example, 3D crystal melting is a first order phase transition, while 2D melting often exhibits two continuous transitions[33]. Similarly, here we found that the $v_a$ ($q_1$) curve at the crystal-to-glass transition is continuous in 2D (Fig. 1f), but makes an abrupt jump in 3D (Fig. 4c), which is consistent with the behaviors of other quantities in ref. [16].

Besides the FPLs in reciprocal space, we discovered power laws about the medium-range $g(r)$ peaks. Although $g(r)$ has been intensively studied in liquids, crystals and glasses, the shift of the $j$th peak has rarely been explored because (1) the positions of medium-range peaks are difficult to measure accurately from the Fourier transform of $S(q)$ in scattering experiments, and (2) the trivial relation, $v_a \propto 1/\phi \propto r_j^D$, is expected to hold. Surprisingly, we find that $d_j$ can deviate from $D$ (Fig. 6f) when the pair interaction has two length scales. $d_j < D$ in SH systems and $d_j = D$ in HH and WCA systems. These real-space results are similar to those in reciprocal space, suggesting that they have the same structural origin.

The results bring new insights on material fabrication. For example, how particle interaction affects material properties is a key question in materials science, but the understanding is limited. For instance, soft solvent particles are empirically argued to be responsible for the elastic modulus of metallic glasses[35]. Here we found that two length scales in the pair potential result in $d \neq D$, indicating that the soft particles play a key role in the FPL and the structural change in glasses. Our results predict that compressing metallic glasses composed of hard-sphere-like atoms will result in $d = D$, and the higher the fraction of soft atoms like Ce, the more $d$ deviates from $D$. Fabricating ultrafine-grained polycrystals is another important challenge in materials science as they are unstable and tend to coalesce into larger grains[36]. We found that a large $\eta$, i.e. more soft particles and SS bonds, causes a broad crossover regime, which corresponds to a broader regime of ultrafine-grained polycrystals with inverse-Hall–Petch behavior and abnormally large compressibility. The fraction of the soft shells of soft particles determines $d$. 2DSH glasses can only form at $1 < d < 2$. Beyond this range, the systems can only form large-grained polycrystals. These results provide guidance for fabricating ultrafine-grained polycrystals and glasses with different degrees of nonuniform deformation under compression.

## Methods

**Simulation methods.** We performed Brownian dynamics simulations for 2DWCA and 3DWCA systems using LAMMPS[37] and event-driven molecular dynamics simulations[38] for the other four types of systems. All the simulations were performed under periodic boundary conditions in $NVT$ ensembles. Samples were relaxed for long enough at each $\phi$.

**Simulations of soft/hard and hard/hard mixtures.** 2DSH: The simulations contained $N = 12,800$ disks with the mixing ratio $\eta = N_S/N$, where $N_S$ is the number of soft disks. The soft particles had square-shoulder potential (Fig. 1e)

$$U(r) = \begin{cases} \infty, & r \leq \sigma \\ U_0, & \sigma < r \leq \lambda\sigma \\ 0, & \lambda\sigma < r \end{cases} \qquad (8)$$

where $\sigma$ and $\lambda\sigma$ are the diameters of the inner core and outer shell, respectively. $\sigma$ serves as the length unit. $U_0$ is the height of the shoulder. The pair potential of the hard particles

$$U(r) = \begin{cases} \infty, & r \leq \lambda\sigma \\ 0, & \lambda\sigma < r \end{cases}. \qquad (9)$$

The packing fraction was calculated as

$$\phi = \frac{N}{A}\frac{\pi\sigma^2}{4}\left((1-\eta)\times\lambda^2 + \eta\times 1^2\right), \qquad (10)$$

where $A$ is the area of the box.

Particles were randomly distributed in the box at $\phi = 0.62$, and then relaxed at $T = 2.0\ U_0/k_B$ for a time period of $10^5 t_0$ and finally equilibrated at $T = 0.133U_0/k_B$ for $10^5 t_0$. $k_B$ is the Boltzmann constant. $t_0 = \sqrt{m\sigma^2/U_0}$ is the amount of time a disk takes to move a distance $\sigma$, where $m$ is the unit of mass. To facilitate the equilibration, initial velocities of particles were reassigned every $10^4 t_0$ with a Gaussian distribution. Starting from $\phi = 0.62$, crystals were compressed into higher packing fractions using the Lubachevsky–Stillinger algorithm[39]. All the results were measured at $T = 0.133U_0/k_B$, a low temperature used to thermalize the system[16].

The results of the 2DSH system with $(\eta, \lambda) = (0.5, 1.3)$ are shown in Figs. 1f, 2, 5b and 6c. Other values of $(\eta,\lambda)$ were explored and some of them are shown in Fig. 3. $\eta$ varies from 0.35 to 0.60 in Fig. 3a, in which range the system can be compressed to the glass state[16]. During compression, defects steadily accumulated through collapse of shoulders in soft particles (Fig. 5a), which caused the crystal to transform into glass (Supplementary Fig. 1). As a nonequilibrium state, glass depends not only on state parameters such as temperature, density and pressure, but also on its fabrication history[2]. We compared two glass states compressed from a liquid (Fig. 3a, b) and from a crystal (Fig. 1f) at the same $(\eta, \lambda) = (0.5, 1.3)$, and found similar FPLs: $d = 1.36$ in Fig. 1f and $d = 1.35$ in Fig. 3a, indicating that the FPL is insensitive to the glass formation pathway.

3DSH: The simulations were performed in a cubic box containing 5000 soft and 5000 hard spheres. The initial state was set to a fluid with $\phi = 0.3$ and relaxed at $T = 2.0U_0/k_B$ to obtain different configurations across trials. Then it was directly compressed into the target $\phi$ (Fig. 4d) and relaxed at $T = 0.2U_0/k_B$ for $10^5 t_0$. The packing fraction is defined as

$$\phi = \frac{N}{V}\frac{\pi\sigma^3}{6}\left((1-\eta)\times\lambda^3 + \eta\times 1^3\right). \qquad (11)$$

The system with $(\eta, \lambda) = (0.4, 1.25)$ exhibits similar features at the crystal-glass transition to those of the 2DSH systems[16].

The simulations of 2DHH and 3DHH systems are the same as those of 2DSH and 3DSH systems, except that the binary HH spheres cannot form crystals at a low $\phi$. The time unit for HH systems $t_0 = \sqrt{m\sigma^2/(k_B T)}$.

**Simulations of WCA systems.** WCA potential[15]

$$U(r) = \begin{cases} 4U_0\left\{\left(\frac{\sigma}{r}\right)^{12} - \left(\frac{\sigma}{r}\right)^6 + \frac{1}{4}\right\}, & r \leq 2^{1/6}\sigma \\ 0, & r > 2^{1/6}\sigma \end{cases} \qquad (12)$$

where $\sigma = 1.3$, 1.15 and 1.0 for large-large, large-small and small-small particle interactions, respectively. $U_0 = 1.0$ is the unit of energy. WCA potential is a well-known short-range repulsive potential which has often been used to model colloidal interactions[14,15,40]. At each $\phi$, particles were randomly distributed in the box. After energy minimization using the FIRE algorithm[41], the system was relaxed at $T = 0.002\varepsilon/k_B$ for $10^8$ steps with the time step $\delta t = 0.001$. The packing fraction was calculated using Eq. 10 for the 2D case and Eq. 11 for the 3D case with the effective diameter $2^{1/6}\sigma$.

**Structural identification.** To characterize the local crystalline order of each particle, we used the modified orientational order parameter[16,42]. This parameter is

more accurate than the conventional bond-orientational order parameter because each neighbor is properly weighed by the corresponding edge of the Voronoi cell[42]. For 2D systems, the modified orientational order parameter

$$\psi_{6j} = \frac{1}{l_{tot}} \sum_{k=1}^{N_j} l_{jk} e^{i6\theta_{jk}}, \tag{13}$$

where $\theta_{jk}$ is the orientational angle of the bond between particle $j$ and its neighbor $k$. The Voronoi polygon[43] has $N_j$ edges with perimeter $l_{tot}$, and the length of the edge between $j$ and $k$ is $l_{jk}$. A higher $|\psi_{6j}|$ represents a higher crystalline order. Particles with three or more crystalline bonds were defined as crystalline, where a crystalline bond (between particles $j$ and $k$) is one that satisfies $|\psi_{6j} \cdot \psi_{6k}^*| > 0.6$[16]. Two neighboring crystalline particles belong to the same grain if the difference between their orientational angles $|Arg(\psi_{6j}) - Arg(\psi_{6k})| \le 5.0°$. Noncrystalline particles and single isolated crystalline particles are defined as disordered[16].

For 3D systems, the modified orientational order parameter

$$q_{li} = \sqrt{\frac{4\pi}{2l+1} \sum_{m=-l}^{l} \left| \sum_{j=1}^{n} \frac{A_j}{A} Y_{lm}(\theta_{ij}, \phi_{ij})^2 \right|}, \tag{14}$$

where $\theta_{ij}$ and $\phi_{ij}$ are the spherical angles of the vector from particle $i$ to its $j$th nearest neighbor. $A_j$ is the area of the Voronoi facet to the $j$th neighbor. $A$ is the total surface area of the Voronoi cell. $Y_{lm}$ is a spherical harmonic function of degree $l$ and order $m$. $q_{l=6} \le 0.4$ are disordered particles; and $q_{l=6} > 0.4$ are crystalline particles[44]. Crystalline particles with $q_{l=4} > 0.143$ are defined as having an face-centered cubic (FCC) structure and the rest are hexagonal close-packed (HCP) structures (Fig. 4d)[44].

## Data availability

The data that support the findings of this study are available from the corresponding author upon reasonable request.

## Code availability

The codes that are used to generate results in the paper are available from the corresponding author upon reasonable request.

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

## Acknowledgements

H.Z. thanks Marcus Bannerman, Xunli Wang, Qiaoshi Zeng, Michio Otsuki and Hua Tong for helpful discussions. This work was supported by the RGC GRF (Grants Nos. 16302316, 16301117 and 16302518).

## Author contributions

H.Z. and Y.H. proposed the project. H.Z. performed the simulation and the data analysis. H.Z., K.Q. and Y.H. discussed the results and developed the theoretical explanations. H.Z. and Y.H. wrote the manuscript.

## Competing interests

The authors declare no competing interests.
