## [Peer Review File · Nature Communications]

Reviewers' comments:

Reviewer #1 (Remarks to the Author):

In this paper, Zhang et al. analyse the relation between volume and position of the first diffraction peak for several models of glasses and crystals with distinct pair interactions, dimension $d=2$ and $d=3$. For each model they obtain a power law relation between these two quantities, and having at hand different pair potentials, structures, and space dimensions allows them to draw more conclusions than previous work on specific models.

The paper is in general well written and could be published, except for the following few points: (i) the "thermodynamic model" presented in the manuscript is not well explained, and I could not really follow what is being done here; in particular, the model mixes simple relations with numerical observations, so this is not really a model. It is unclear if this part really illuminates the numerical results. (ii) the authors study simple yet specific models, but they keep referring with more complex results obtained for more complex mixtures (for instance in metallic glasses). It is unclear that their simple models really have anything to say about these complex experimental systems. (iii) the protocol to generate glasses is important, yet is only explained in the SI. The title is misleading since the authors do not really "compress" mechanically the glasses, but rather prepare different glasses at different state points completely independently. It would be useful to explain in the introduction if experiments and simulations all have followed the same protocol, and whether the power law discussed by the author could also depend on the specificities of their chosen protocol.

Reviewer #2 (Remarks to the Author):

This paper reports a comprehensive study of the fractal behavior of 2D and 3D glasses, by way of computer simulations. For each type of glass (2D or 3D), the authors studied three different systems characterized by different interaction potentials, namely hard-hard spheres, WCA potentials, and soft-hard spheres. Through these systematic, numerical studies, the authors not only confirmed the existence of power law, but also uncovered the underlying physics. Power-law behaviors were seen in both 2D and 3D systems, but only for soft-hard spheres. The authors further demonstrated that this was a result of the two length scales in the interaction potential, proposed by Jagla in 1999. This finding has significant implications, because a two-length scale potential also naturally leads to a liquid-liquid-phase-transition, which apparently is the case as Figure 2 shows. Also remarkably, the authors found that for a two-dimensional glass, the fractal dimension D scales with a single parameter, X , which measures the size mismatch allowed during compression.

Further detailed simulations also shed light on some of the lingering puzzles in the study of fractal behavior. For example, a recent MD simulation suggested that $Zr_{46}Cu_{54}$ does not show fractal power law scaling. The authors were able to demonstrate that Zr and Cu behaved more like hard-hard spheres, and thus no fractal behavior is expected. On the other hand, a fractal power law scaling is expected for La- and Ce-based metallic glasses, because electronically, La and Ce atoms are soft and compressible. The authors also analyzed the scaling in real space, and found that the peaks in $g(r)$ at medium-range length scales exhibit a power law of similar dimensionality, suggesting power law scaling have the same structural origin.

This paper addresses an important question in amorphous materials. The presentation is clear and easy to follow. However, there are still a couple of issues which I would like the authors to consider. I would be pleased to endorse the manuscript for publication in Nature Communications if the authors can properly address my questions.

1. A discussion of D values in the cross-over region deserves re-consideration. The transition can be gradual, like $D \sim 4.6$ in a 2D glass, or abrupt, like $D \sim 4.3$ and then nearly zero in a 3D glass. It appears to me that the scaling in the cross-over region describes the nature of the transition from glass to crystalline states, rather than the dimension of a fractal system. What is the physical meaning of $D > 2$ in a 2D glass?
2. Why do the first few peaks in the PDF (for the 2D system) split? Is it due to statistics of the simulation? A power law scaling is not expected at short distances. Still, it would be interesting to analyze the scaling of the first few peaks, if possible.

There are also a few typos and formatting issues.

1. Y-axis labels in Figure 3c and 3d should be rotated by 90-degree.
2. P. 9, 1st paragraph, line 6, ... result (in)

Reviewer #3 (Remarks to the Author):

The manuscript aims at understanding the fractional power law relationship between the average atomic volume and the first diffraction peak position in glasses, which is a phenomenon that has been observed in a number of experimental studies. Through numerical simulations of six models, the authors provide a better understanding on the mechanism of this behavior. In general, the presented data is technically solid, which supports well most of the conclusions. The results are novel and interesting, providing new insights into the understanding of structural change in glasses in response to mechanical deformation such as compressions. I would recommend publication if the following points can be properly addressed.

- 1) Section Thermodynamic model for the FPL: The entropy in glasses contains two parts, the configurational entropy and the vibrational entropy. I believe that the model neglects the configurational part because it does not consider the possibility of change of the configuration during compressions. For example, the free energy could depend on density ϕ , without changing the overlapping areas, if the number of configuration is allowed to change. Neglecting the configurational contribution might be a good approximation in the regime $d = D$ since the behavior is similar to crystals, but it is not clear if the approximation is still good in the situation $d < D$. In any case, calling the model 'thermodynamic' is confusing because in the end, the derivation of eq. (5) is not based on free energy, entropy, or any other thermodynamic quantities.
- 2) Fig. 4c: What is the green regime in 3D? In 2D, the green regime in Fig. 1f corresponds to the inverse-Hall-Petch regime plus shadow glass regime (Fig. 2).
- 3) Fig. 5b-c: How to define a compressed bond? Is it possible to find partially compressed bonds? If yes, are they counted in n_{SS} and n_{HS} ?
- 4) Eq. (5) and Fig. 5d: Why do the coefficients C_{SS} and C_{SH} not depend on ϕ ?
- 5) Section Power laws in real-space: The exponents d_j found in $g(r)$ are all different from d , because $d_j = 1.39 - 1.80$ in Fig. 6f, and for the same set of parameters, $d = 1.35$ in Fig. 3a-b. Therefore, a quantitative agreement is missing. The best agreement is found for $j=8$, which seems to suggest that the long-range correlation, instead of the medium-range order, is relevant.
- 6) Methods: the simulation time unit should be defined for the HH models.

7) Methods: 'FIRA algorithm' should be 'FIRE algorithm'.

8) Fig. S6 caption: SS bonds are not red circles and SH bonds are not blue squares.

9) Fig. S5c should appear after Figs. S6 and S7 in the SI.

Reviewer #1 (Remarks to the Author):

In this paper, Zhang et al. analyse the relation between volume and position of the first diffraction peak for several models of glasses and crystals with distinct pair interactions, dimension $d=2$ and $d=3$. For each model they obtain a power law relation between these two quantities, and having at hand different pair potentials, structures, and space dimensions allows them to draw more conclusions than previous work on specific models.

The paper is in general well written and could be published, except for the following few points: (i) *the "thermodynamic model" presented in the manuscript is not well explained, and I could not really follow what is being done here; in particular, the model mixes simple relations with numerical observations, so this is not really a model. It is unclear if this part really illuminates the numerical results.*

Reply: We have revised "thermodynamic model" to "theoretical explanation" in the section title. In this section, we explain the observations of the simulations by two approaches: 1, How effective for compressing a SS and a SH bond on lowering the entropy (the last paragraph of page 4) and on inducing nonuniform local deformation (paragraph 1 of page 6). 2, Using equations 4 and 5 and simulation results ($C_{SS} < 0$ and $C_{SH} > 0$) to explain the observations.

To clarify the structure and results in this section, we add a new paragraph in page 6: "The above arguments about the SS- and SH-bond compressions without using any simulation results have qualitatively explained $d > D$ in the crossover regime and $d < D$ in the glass regime, and $d(\lambda)$ behaviors. The simulation results in this section just provide consistency checks and are not necessary for the theoretical explanations. Beside the above approach, next we introduce another approach based on Eqs. 4 and 5 below. This second approach needs simulation results to fully explain the observations. Thus the first approach above provides a full qualitative explanation, while the second approach below is just a consistency check."

In order to highlight the key method in the theoretical analysis, we change "The results are explained by random close packing, bond deformation and a thermodynamic model" to "The results are explained by two types of bond deformations" in the abstract.

(ii) *the authors study simple yet specific models, but they keep referring with more complex results obtained for more complex mixtures (for instance in metallic glasses). It is unclear that their simple models really has anything to say about these complex experimental systems.*

Reply: It is well accepted that some typical metallic glasses such as Zr-Cu and Zr-Ni alloys can be modeled by hard spheres. Our $d=3$ in 3DHH system is consistent with the reported $d=3$ for Zr₄₆Cu₅₄ alloy [9], see paragraph 5 in page 8. In page 8 paragraph 6, after "Ce atoms are much softer than other atoms [22, 29] and can be described as spheres with a square-shoulder potential [17]", we add "because its 4f-electron orbit is localized at low pressure and delocalized at high pressure, resulting in atomic volume collapse [3,22]". In addition, the glass-glass transitions are observed in both Ce-based alloys [3] and our soft/hard mixtures [15], which further confirms the similarity between these systems, see the last paragraph of the section "Power laws in 2D systems".

(iii) the protocol to generate glasses is important, yet is only explained in the SI. The title is misleading since the authors do not really "compress" mechanically the glasses, but rather prepare different glasses at different state points completely independently. It would be useful to explain in the introduction if experiments and simulations all have followed the same protocol, and whether the power law discussed by the author could also depend on the specificities of their chosen protocol.

Reply: We thank the reviewer for the suggestion, and have changed the title to "Power laws in pressure-induced structural change of glasses". In page 2 paragraph 2, we briefly explained the protocol "Each state is directly compressed from a low-density liquid (see Methods). Hence states under different pressures are uncorrelated and the structural FPLs are not related to affine or non-affine deformation. As a comparison, we also compressed the 2DSH system step by step and obtained the same FPLs." In the resubmission, we further add the following text right after the above text "The step by step compression in 3DSH system yields similar FPLs in the crystal and glass regimes, but different behavior in the crossover regime." We also add the following text near the end of the section "Power laws in 3D systems": "If the 3DSH system is compressed step by step, the crystal behavior of the power law extends to the crossover regime and exhibits a jump at the onset of the glass regime. Such protocol dependence in the crossover regime should be due to the first-order-like polycrystal-glass transition in 3D."

Reviewer #2 (Remarks to the Author):

This paper reports a comprehensive study of the fractal behavior of 2D and 3D glasses, by way of computer simulations. For each type of glass (2D or 3D), the authors studied three different systems characterized by different interaction potentials, namely hard-hard spheres, WCA potentials, and soft-hard spheres. Through these systematic, numerical studies, the authors not only confirmed the existence of power law, but also uncovered the underlying physics. Power-law behaviors were seen in both 2D and 3D systems, but only for soft-hard spheres. The authors further demonstrated that this was a result of the two length scales in the interaction potential, proposed by Jagla in 1999. This finding has significant implications, because a two-length scale potential also naturally leads to a liquid-liquid-phase-transition, which apparently is the case as Figure 2 shows. Also remarkably, the authors found that for a two-dimensional glass, the fractal dimension D scales with a single parameter, X , which measures the size mismatch allowed during compression.

Further detailed simulations also shed light on some of the lingering puzzles in the study of fractal behavior. For example, a recent MD simulation suggested that Zr₄₆Cu₅₄ does not show fractal power law scaling. The authors were able to demonstrate that Zr and Cu behaved more like hard-hard spheres, and thus no fractal behavior is expected. On the other hand, a fractal power law scaling is expected for La- and Ce-based metallic glasses, because electronically, La and Ce atoms are soft and compressible. The authors also analyzed the scaling in real space, and found that the peaks in $g(r)$ at medium-range length scales exhibit a power law of similar dimensionality, suggesting power law scaling have the same structural origin.

This paper addresses an important question in amorphous materials. The presentation is clear and easy to follow. However, there are still a couple of issues which I would like the authors to consider. I would be pleased to endorse the manuscript for publication in Nature Communications if the authors can properly address my questions.

1. A discussion of D values in the cross-over region deserves re-consideration. The transition can be gradual, like $D \sim 4.6$ in a 2D glass, or abrupt, like $D \sim 4.3$ and then nearly zero in a 3D glass. It appears to me that the scaling in the cross-over region describes the nature of the transition from glass to crystalline states, rather than the dimension of a fractal system. What is the physical meaning of $D > 2$ in a 2D glass?

Reply: In the crossover regime, compressing one large-lambda SS bond or multiple small-lambda SS bonds are equivalent in changing the area fraction, but different in changing q_1 because the latter case has a more uniform structure change. Hence d is larger for a softer particle systems; $d=D$ for the least soft system, i.e. binary HH systems. Hence $d > D$ for SH systems. This argument can be found in paragraph 1 of page 6 and the answer of question 3 in Discussion section.

Page 5 paragraph 1 shows that d deviates from D oppositely in the crossover and glass regimes because SS and SH bonds are obviously anticorrelated in space. Since $d > D$ in the crossover regime as we explained above, $d < D$ in the glass regime. This explanation did not assume the existence of any fractal structure. In fact, $d > D$ indicates the absence of any fractal structure in a D -dimensional space. We add “This explanation for $d < D$ and $d > D$ in different regimes does not need the assumption of any fractal structure. In fact, fractal structures with dimension $d > D$ cannot exist in a D -dimensional space.” at the end of the second paragraph in the answer of question 3 in Discussion section.

2. Why do the first few peaks in the PDF (for the 2D system) split? Is it due to statistics of the simulation? A power law scaling is not expected at short distances. Still, it would be interesting to analyze the scaling of the first few peaks, if possible.

Reply: The split is due to the different sizes of particles, not due to statistics. At the end of page 7, we revised the text as “For binary systems, the first peak of $g(r)$ splits into three subpeaks because the first-layer neighbors have three typical separations corresponding to S-S, S-H and H-H bonds (Supplementary Fig.~S5c). These three subpeak positions cannot deviate much from their corresponding bond lengths, thus r_1 changes very little, resulting exceptionally large exponents in the power law (Supplementary Fig.~S8). The second- and third-layer peaks of $g(r)$ split into more subpeaks which interferences with each other, thus their ambiguous peak positions are not measured. We focus on the positions of the unimodal peaks, ...”

There are also a few typos and formatting issues.

1. Y-axis labels in Figure 3c and 3d should be rotated by 90-degree.

Reply: Thank you. It has been corrected.

2. P. 9, 1st paragraph, line 6, ... result (in)

Reply: It has been corrected.

Reviewer #3 (Remarks to the Author):

The manuscript aims at understanding the fractional power law relationship between the average atomic volume and the first diffraction peak position in glasses, which is a phenomenon that has been observed in a number of experimental studies. Through numerical simulations of six models, the authors provide a better understanding on the mechanism of this behavior. In general, the presented data is technically solid, which supports well most of the conclusions. The results are novel and interesting, providing new insights into the understanding of structural change in glasses in response to mechanical deformation such as compressions. I would recommend publication if the following points can be properly addressed.

1) *Section Thermodynamic model for the FPL: The entropy in glasses contains two parts, the configurational entropy and the vibrational entropy. I believe that the model neglects the configurational part because it does not consider the possibility of change of the configuration during compressions. For example, the free energy could depend on density ϕ , without changing the overlapping areas, if the number of configuration is allowed to change. Neglecting the configurational contribution might be a good approximation in the regime $d=D$ since the behavior is similar to crystals, but it is not clear if the approximation is still good in the situation $d < D$. In any case, calling the model 'thermodynamic' is confusing because in the end, the derivation of eq. (5) is not based on free energy, entropy, or any other thermodynamic quantities.*

Reply: We add the following text in the first paragraph of page 5 “**Note that the configurational entropy is neglected because we focus on the entropy change instead of entropy when a bond is compressed in a given glassy configuration.**” Moreover, our glasses are compressed from crystals, thus are well below the melting point. The configurational entropy in such a deep glassy state could be negligible since the configurational entropy is zero at the ideal glass transition point.

We have revised "thermodynamic model" to “**theoretical explanation**” in the section title.

2) *Fig. 4c: What is the green regime in 3D? In 2D, the green regime in Fig. 1f corresponds to the inverse-Hall-Petch regime plus shadow glass regime (Fig. 2).*

Reply: The green regime should correspond to the shadow glass regime only, without the inverse-Hall-Petch regime for a polycrystal because the structure is totally disorder in this regime as shown in Fig.4d.

3) Fig. 5b-c: How to define a compressed bond? Is it possible to find partially compressed bonds? If yes, are they counted in n_{SS} and n_{HS} ?

Reply: We add the following text “The compressed SS and SH bonds are defined as pairs belonging to the first and second subpeaks of the first-layer neighbor separations, respectively, see Supplementary Fig. S5c.” in Fig.5a caption; and “The red and blue regions in \mathbf{c} define the compressed SS bond at $r < 1.14$, the compressed SH bond at $1.14 < r < 1.28$ and the uncompressed bonds at $1.28 < r < 1.5$. Once the soft shoulder is compressed, the pair distance tends to be as close as it could be, i.e. the bond is fully compressed and the inner hard core is in contact with the neighboring particle. Such configuration yields the largest free volume, thus are more favorable in entropy.” in Fig. S5 caption.

Yes, we counted partially compressed bonds in n_{SS} and n_{HS} .

4) Eq. (5) and Fig. 5d: Why do the coefficients C_{SS} and C_{SH} not depend on ϕ ?

Reply: In page 6, C_{SS} (or SH) is defined as the partial derivative at fixed ϕ and n_{SH} (or SS), thus it does not depend on ϕ . In fact, the influence of ϕ is included in n_{SS} and n_{SH} while C_{SS} and C_{SH} are the coefficients depending on λ .

5) Section Power laws in real-space: The exponents d_j found in $g(r)$ are all different from d , because $d_j = 1.39 - 1.80$ in Fig. 6f, and for the same set of parameters, $d = 1.35$ in Fig. 3a-b. Therefore, a quantitative agreement is missing. The best agreement is found for $j=8$, which seems to suggest that the long-range correlation, instead of the medium-range order, is relevant.

Reply: We only observed that when a system has $d=D$ in reciprocal space, then its d_j also equals to D in real space; when a system has $d < D$ in reciprocal space, then its d_j is also less than D in real space. We did not figure out any mechanism which can produce a quantitative relation between d_j and d . We agree that $d_j=8$ is closer to d in the reciprocal space and deviates from the space dimension D most. On the other hand, the 8th peak is weaker than the 4th peak. Thus it is unclear if the 8th peak or 4th peak contributes more to the structural origin of the FPL. We add “Whether d_j from real space and d from reciprocal space have a quantitative relation is worth to explore in the future.” at the end of section “Power laws in real-space $g(r)$ ”.

6) Methods: the simulation time unit should be defined for the HH models.

Reply: We have added it at the last sentence of the 2nd paragraph in Page 10 “The time unit for HH systems $t_0 = \sqrt{m\sigma^2/(k_{\text{B}}T)}$.”

7) Methods: ‘FIRA algorithm’ should be ‘FIRE algorithm’.

Reply: Thank you. We have corrected it.

8) *Fig. S6 caption: SS bonds are not red circles and SH bonds are not blue squares.*

Reply: Thank you. We have corrected it.

9) *Fig. S5c should appear after Figs. S6 and S7 in the SI.*

Reply: We cite Fig. S5c before Fig. S6 in the resubmission.

Reviewers' comments:

Reviewer #1 (Remarks to the Author):

I think the authors did their best to address the various questions and remarks. I suggest publication.

Reviewer #2 (Remarks to the Author):

The authors satisfactorily answered my question #2, but only partially for question #1. I do not consider $d > D$ to be physically meaningful unless there is a higher dimension which we have not realized. Nevertheless, I am happy to leave this debate to future studies.

I also agree with Reviewer #1 that the original title was not accurate. The revised title is better.

Reviewer #3 (Remarks to the Author):

The authors have addressed most of my questions, expect for the following one.

"4) Eq. (5) and Fig. 5d: Why do the coefficients C_{SS} and C_{SH} not depend on ϕ ?

Reply: In page 6, C_{SS} (or C_{SH}) is defined as the partial derivative at fixed ϕ and n_{SH} (or n_{SS}), thus it does not depend on ϕ . In fact, the influence of ϕ is included in n_{SS} and n_{SH} while C_{SS} and C_{SH} are the coefficients depending on."

The reply does not answer my question. For example, if $F(x,y)=xy$, then $\frac{\partial F}{\partial x} = y$, which depends y . The independence of C_{SS}/C_{SH} on ϕ could be a good approximation, but I do not see any supportive data or justification from the physics.

Reviewer #3 (Remarks to the Author):

The authors have addressed most of my questions, expect for the following one.

"4) Eq. (5) and Fig. 5d: Why do the coefficients C_{SS} and C_{SH} not depend on ϕ ?

Reply: In page 6, C_{SS} (or SH) is defined as the partial derivative at fixed ϕ and n_{SH} (or SS), thus it does not depend on ϕ . In fact, the influence of ϕ is included in n_{SS} and n_{SH} while C_{SS} and C_{SH} are the coefficients depending on."

The reply does not answer my question. For example, if $F(x,y)=xy$, then $\frac{\partial F}{\partial x} = y$, which depends y . The independence of C_{SS}/C_{SH} on ϕ could be a good approximation, but I do not see any supportive data or justification from the physics.

Reply: We agree with the reviewer that our previous reply does not rule out the case like $f(x,y)=xy$. For our function $q_1(\phi, n_{SS}, n_{SH})$, the three variables all represent the volume changes, just from three different mechanisms; please see page 6, paragraph 3 "In 2DSH systems, the volume change arises from the squeezing of the free volume characterized by ϕ , and the compression of the SS and SH bonds." Therefore we expect $f(x,y,z)$ is something like $x+y+z$ rather than $x*y*z$, although we do not have a rigorous proof. It is our assumption based on the observations from Fig.S6 without a deeper physical explanation. Fig.s6 shows that $\ln(q_1)$ is linear to the concentration which is linear to volume fraction, (please see page 6 " $\ln \tilde{q}_1$ is proportional to n_{SS} and n_{SH} in Supplementary Figure~6a, and we find a similar relationship in $\text{Cu}_c\text{Zr}_{1-c}$ using the data in ref.~\cite{calvayrac83} (Supplementary Figure~6b). Consequently, $\frac{\partial \ln \tilde{q}_1}{\partial n_{SS}} \bigg|_{\phi, n_{SH}}$ and $\frac{\partial \ln \tilde{q}_1}{\partial n_{SH}} \bigg|_{\phi, n_{SS}}$ are denoted by two constants C_{SS} and C_{SH} respectively.")

In the previous resubmission, we highlighted that this approached needs simulation results and thus is not a full explanation: page 6 paragraph 2 "Beside the above approach, next we introduce another approach based on Eqs.~4 and 5 below. This second approach needs simulation results to fully explain the observations. Thus the first approach above provides a full qualitative explanation, while the second approach below is just a consistency check".

REVIEWERS' COMMENTS:

Reviewer #3 (Remarks to the Author):

The authors have addressed my question. I recommend the manuscript for publication.